# Muscle anatomy is reflected in the spatial organization of the spinal motoneuron pools

Rachel I. Taitano [1], Sergiy Yakovenko[1] & Valeriya Gritsenko [1✉]

Neural circuits embed limb dynamics for motor control and sensorimotor integration. The somatotopic organization of motoneuron pools in the spinal cord may support these computations. Here, we tested if the spatial organization of motoneurons is related to the musculoskeletal anatomy. We created a 3D model of motoneuron locations within macaque spinal cord and compared the spatial distribution of motoneurons to the anatomical organization of the muscles they innervate. We demonstrated that the spatial distribution of motoneuron pools innervating the upper limb and the anatomical relationships between the muscles they innervate were similar between macaque and human species. Using comparative analysis, we found that the distances between motoneuron pools innervating synergistic muscles were the shortest, followed by those innervating antagonistic muscles. Such spatial organization can support the co-activation of synergistic muscles and reciprocal inhibition of antagonistic muscles. The spatial distribution of motoneurons may play an important role in embedding musculoskeletal dynamics.

[1] West Virginia University, Morgantown, WV, USA.  ✉email: vgritsenko@mix.wvu.edu

The spinal cord drives the musculoskeletal system by rapidly integrating ongoing execution-related (efferent) signals with proprioceptive (afferent) signals to enable motor function in presence of uncertain environmental conditions. The afferent signals are somatotopically organized carrying information from different body parts to specific spinal cord segments[1,2]. This broadly agrees with the spatial distribution of MN pools innervating the corresponding body parts. Specifically, the consensus is that the MN pools innervating muscles of the lower limb that have synergistic function at a given joint are located closer together than those that have antagonistic function about that joint[3,4]. Furthermore, both the primary sensory and motor cortices that process the afferent information and generate efferent output are somatotopically organized as famously represented by the sensory and motor homunculi, respectively[5–8]. This organization can effectively embed the anatomical and dynamical properties of the controlled body parts in the neural substrate, a concept referred to as internal models[9–12]. It is unknown how much the spinal cord contributes to the neural embedding of muscle anatomy, especially for the arm with its complex musculature that supports a wide repertoire of movements.

A single motoneuron (MN) pool is a group of ventral horn neurons that innervate a single muscle. Each MN pool spans multiple segments forming a long narrow column. Neurodevelopmental studies have shown that the relative location of MN pools within the lateral motor column correlates with embryonic origin of the muscles they innervate[13–15]. For the upper limb, the two muscle precursors, the dorsal and ventral muscle masses, give rise to the MN pools whose axons form nerves that innervate dorsal and vernal musculature[13,15]. The organization of MN pools innervating the hindlimb/lower limb is generally preserved across vertebrates[2,16–18], including humans[19–22]. This is not entirely unexpected as the function of the hindlimb/lower limb is largely conserved to perform stereotypical movements such as locomotion, scratching, or swimming. However, the human arm and hand has a large repertoire of both gross stereotypical movements and highly dexterous movements. Yet the detailed anatomical data about the distribution of human MN pools innervating the arm and hand is sparse. The knowledge of the rostrocaudal distribution of MN pools innervating the human arm muscles was derived from the anatomy of spinal roots and peripheral nerves as summarized in the human anatomy textbooks, e.g.[23]. However, the organization of the human MN pools in the transverse plane is less clear, as the methods used to map it lack the identification of their innervation targets[20]. Macaque and human upper limbs can perform similar functions. Therefore, comparing the locations of macaque MNs that innervate upper limb muscles with the anatomical relationships between these muscles in macaques and humans can help us understand the meaning of the spatial organization of the MN pools in the spinal cord.

Here we used the detailed anatomical organization of MN pools innervating the upper limb muscles of macaques obtained with retrograde staining[24] to create a spatial model of the MN distributions in the spinal cord ventral horn. This approach was inspired by the spatial model of the lumbosacral MNs innervating hindlimb muscles of the cat[18]. We also took advantage of the detailed information about the organization of the musculoskeletal system developed using biomechanical modeling techniques[25–27]. Our goal is to determine how the anatomical substrate in the spinal cord supports neural embedding of muscle anatomy. We will test the hypothesis that the organization of MN pools embeds the musculoskeletal organization.

## Results
**Rostrocaudal distribution of MN pools**. The MN pools in the cervical enlargement of macaques were located in the lateral part

of the ventral horn (Fig. 1a) and distributed along multiple segments (Fig. 1b, c). This organization is similar to the motoneurons in the lumbosacral enlargement of cats[18] and humans[20,23] innervating the hindlimbs/legs. The cervical motoneurons labeled in the Jenny and Inukai study[24] innervated muscles located distally on the upper limb of the macaque (Fig. 2a). The anatomical arrangement of this musculature closely follows that of human forearm and hand (Fig. 2b). MNs innervating proximal muscles in the macaque, such as the biceps (BIC) and triceps (TRI), together with flexor and extensor carpi radialis muscles (FCR and ECR respectively) were in the rostral part of the cervical enlargement, while the rest of the MN pools were in the caudal cervical and thoracic segments (Fig. 1c). In humans, the rostro-caudal distribution of MN pools has been estimated from the distribution of ventral roots across spinal segments and the muscles they innervate[23]. The rostrocaudal distributions of human and macaque MN pools overlapped for all muscles (Fig. 3a). The most prominent differences were for MN pools innervating extensor digitorum communis and superficialis (EDC and EDS respectively), extensor carpi ulnaris (ECU), extensor pollicis longus (EPL), abductor pollicis longus and brevis (APL and APB respectively), and lumbrical (LUM) muscles. They were shifted more rostrally, and some spanned more segments in humans compared to macaques. However, the non-overlapping segments tended to have fewer MNs. The shift in MN pool distributions may be due to the fewer number of spinal segments in humans compared to macaques[17], which may allow for more separation between MN pools in the rostrocaudal dimension in macaques.

**Transverse distribution of MN pools**. The distribution of MN pools in the transverse plane in macaque and human spinal cords also overlapped (Fig. 3b) and was estimated using postmortem staining by cresyl violet, as summarized in Routal and Pal study[20]. The human MN pools in the lateral division of the ventral gray horn overlapped with those in the macaque (Fig. 3b). Although it was not possible to identify which muscles the labeled human MNs innervated, the overlap with the macaque data suggests that the human MN pool labeled as 5 may innervate BIC and the human MN pool labeled as 7 may innervate TRI (Fig. 3b). The non-overlapping human MN pools labeled as 1 and 2 likely innervate more proximal muscles not included in the Jenny and Inukai study[24]. Furthermore, the Routal and Pal study[20] has suggested that the rostrocaudal profiles of individual MN pools split and branch when spanning spinal cord segments (Fig. 1 in ref. [20]). However, neither the macaque MN pools studied here, nor other vertebrate MN pools labeled using the retrograde transport technique from single muscles showed this branching property. This puts the validity of the numbering method used to trace individual MN pools across the segments in the Routal and Pal study[20] in doubt. Overall, this shows that there is some overlap between human and macaque MN pools in the transverse plane. Further studies are needed to identify discrepancies between the locations of MN pools of humans and macaques that cannot be influenced by the differences in methodology used to identify MN pools.

**Spatial analysis of MN pools**. To quantify the anatomical organization of spinal MN pools, we calculated the Euclidian distances between their centers. The MN pools innervating APB, adductor pollicis (AP), dorsal interossei (DIO), and LUM muscles were co-localized as evidenced by the relatively short distances between them (Fig. 4a, b, dark red cluster in the bottom right corner). Other MN pools innervating obviously synergistic muscle pairs were co-localized, such as the MN pool of flexor digitorum superficialis

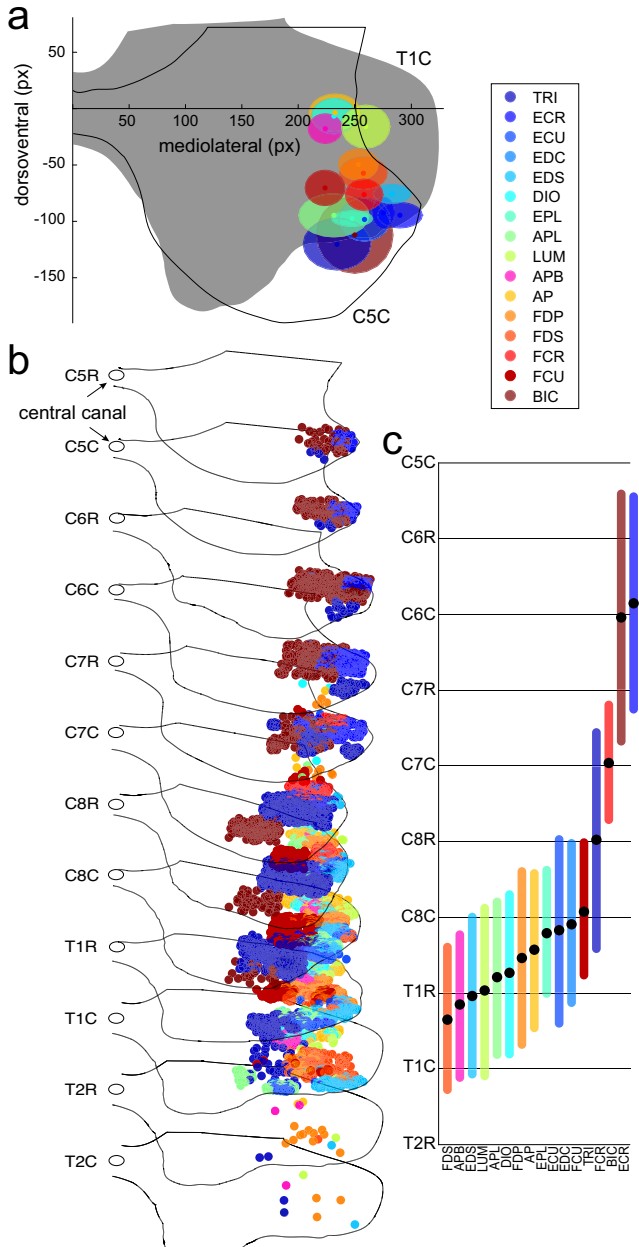

**Fig. 1 Model of motoneuron pool anatomy in a macaque. a** Circles show averaged locations in the transverse plane of the MNs shown in **b**, the shaded areas are standard deviations along the dorsoventral and mediolateral directions in the transverse plane calculated across all motoneuron locations in four cervical and two thoracic segments. Gray matter outlines are shown for the caudal portion of the fifth cervical segment (C5C) and the caudal portion of the first thoracic segment (T1C, gray fill). The axis's origin is at the central canal. **b** 3D view of the individual motoneuron locations (colored circles, circle size is not to scale) and gray matter outlines for spinal segments C5 - T2. Central canal locations are shown as open circles. Motoneuron pool colors are the same as in **a**. The rostrocaudal spacing is not to scale. **c** The rostrocaudal distributions of the motoneuron locations shown in **b**. Black circles indicate averages and colored vertical lines indicate standard deviations across the rostrocaudal coordinate (Z) of MNs shown in **b**.

and profundus (FDS and FDP respectively); EPL and APL; EDC and ECR (Fig. 4a, b). The MN pools innervating less obvious agonists, such as BIC and ECR muscles that would often work together during reaching with hand pronated, were also closely

located in the spinal cord (Fig. 4a, b, dark red cluster in the top left corner). This anatomical arrangement suggests that the MN pools innervating agonistic muscles that perform a single function, such as spreading fingers apart in the case of the first cluster, are located closer together than the MN pools innervating antagonistic muscles that perform distinct functions. We have previously conducted a rigorous analysis of the agonistic and antagonistic relationships between muscles using a model of musculoskeletal anatomy of the human arm[25]. This analysis has been repeated here on the model of the macaque forelimb (Fig. 2a). It has shown that the synergistic muscles whose length changes together (Fig. 4c, high positive correlations) also have short distances between their MN pools. However, excluding the rostrocaudal distribution from the distance calculations shortened the distances between MN pools innervating some antagonistic muscles. For example, in the transverse plane the MN pools of ECR and APL were closely located to the MN pool of antagonistic BIC or the MN pool of EDS was closely located to the MN pool of antagonistic FCR (Fig. 4b). This suggests that the proximity of MN pools innervating antagonists may support the flow of afferent information for reciprocal interactions between these muscles.

**Musculoskeletal anatomy of upper limb muscles**. To compare the anatomical relationships between muscles of macaques and humans, we correlated changes in muscle lengths across the whole range of possible upper limb postures using two musculoskeletal models (Fig. 2). The logic here is that muscle length changes that are positively correlated indicate agonistic relationship, while muscle length changes that are negatively correlated indicate antagonistic relationship. For example, during wrist extension ECU and EDC shorten and their lengths are positively correlated identifying them as agonists, while FDP and FDS lengthen together, identifying them as agonists, and they exhibit negative correlation with the extensor muscles, identifying them as antagonists to extensor muscles. Indeed, all muscles but DIO and LUM had strong agonistic or antagonistic relationships with at least one other muscle based on the length comparisons using the macaque musculoskeletal model (Fig. 4c). This shows that there are strong anatomical relationships between most muscles of the upper limb that are innervated by the MN pools included in the spinal cord model. Moreover, the anatomical relationships between all major muscles of the upper limb of the macaque were very similar to those observed in the human arm (Fig. 5). This suggests that not only the neural anatomy is preserved across species, but also the anatomical relationships between muscles are preserved across species that perform similar movements.

**Comparative analysis**. We examined how the anatomical relationship between muscles is related to the spatial relationship between MN pools innervating these muscles. The Euclidian distances between MN pool centers were not normally distributed, which became even more evident when the rostrocaudal distribution of elongated MN pools was excluded from MN pool center calculations (Fig. 6a, b). The MN pools innervate muscles that have both agonistic and antagonistic relationships as shown by positive and negative correlations between their length changes (Fig. 4). Therefore, when the distances were plotted against the significant muscle length correlations, some relationships between neural structure and muscle anatomy emerged. Firstly, there were much fewer significant anatomical relationships between muscle pairs whose MN pools were the furthest away from each other, $N = 7$ & 3 for the 3D and 2D distances, respectively, out of 29 total significant relationships between muscles they innervate (Figs. 4c and 6c, d; few values above mean distance). Secondly, MN pools that innervate muscles through a

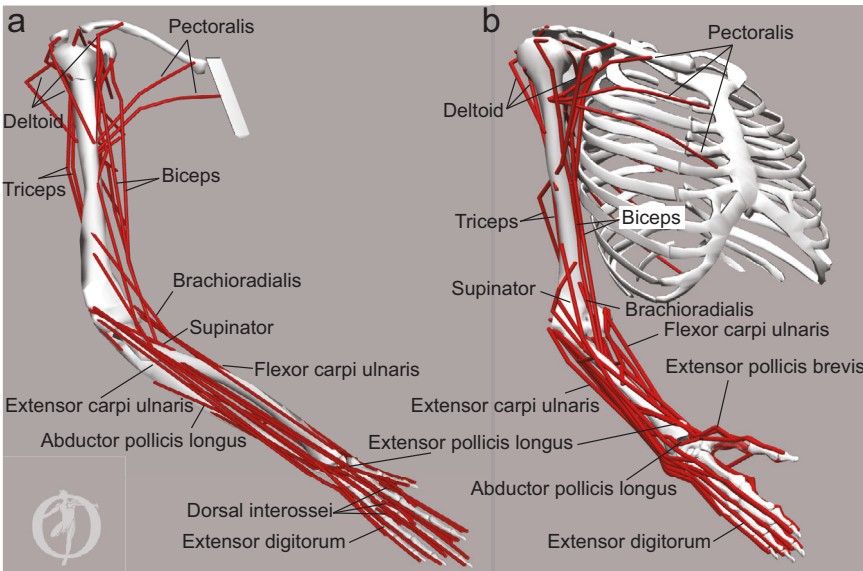

**Fig. 2 Musculoskeletal models.** Screenshots of OpenSim musculoskeletal models of macaque (**a**) and human (**b**) upper limbs. Red lines illustrate the origin, insertion, and wrapping geometry of the musculotendinous actuators representing the anatomical arrangement of individual muscles or muscle compartments. For example, two heads of biceps are modeled as two actuators with origin locations on different bones and a common insertion location. All actuators are shown, but not all are labeled for clarity. The bones are shown for illustration purposes only, the inertial geometries of limb segments to which the actuators attach are not shown. The OpenSim logo is added when exporting screenshots from the graphical user interface.

common nerve tended to be the closest together. For example, the blue dots in the bottom right corner of each plot show distances between MN pools that contribute to the radial nerve and innervate ECU, EDS, EDC, APL and EPL, all extensor muscles with positive correlations between their lengths (Fig. 6c, d; bottom right corner of each plot). Similarly, the two red dots in the bottom right corner of each plot show distances between MN pools that contribute to the median and ulnar nerves through the medial cord of the brachial plexus and innervate APB, AP, FDP, and FDS muscles that move digits synergistically with positive correlations between their lengths. This supports the established view that the MN pools that share the same developmental origins co-localize in the spinal cord.

Lastly, the MN pools that a further away from each other but still below the mean distance value innervate muscles with negative correlations between their lengths, $N = 22$ & 26 for the 3D and 2D distances, respectively, out of 29 total significant relationships between muscles they innervate (Fig. 6c, d; magenta dots on the left side of vertical line). Since the MNs innervate primarily synergistic muscles through the shared nerves, all antagonistic relationships between muscle lengths represent MN pools that innervate them through different nerves. This arrangement is not reflected in the developmental origins of the MN pools. However, the MN pools innervating agonistic muscles tended to be closer together and surrounded by the MN pools innervating antagonistic muscles. There was a significant linear relationship between the R values and the 2D distances between MN pools (Fig. 6d). The lack of significant relationship between the R values and the 3D distances is likely due to the limited distribution of MN pool centers along the rostrocaudal direction. However, the null hypothesis of no difference between distances associated with positive and negative R was rejected for the 3D distances ($D(27) = 0.51$, $p = 0.015$), but not for the 2D distances ($D(29) = 0.41$, $p = 0.081$). These results support our hypothesis that the structural spinal organization of MN pools embeds the musculoskeletal organization. Overall, our results suggest that the MN pools are positioned to support the reciprocal interactions between the muscles they innervate.

## Discussion

Here, we have developed the first spatial model of cervico-thoracic MN distributions in the macaque. We show that the gross MN organization agrees with findings from previous studies in the cat and mouse lumbosacral plexus[4,16,28] and the developmental studies in birds[13]. Using our model, we have shown that the MN distributions in the macaque spinal cord are broadly similar to those in human spinal cord (Fig. 3). Using the musculoskeletal models of the macaque and human upper limbs, we have also shown that the muscle anatomy of both species serves similar agonistic and antagonistic actions (Fig. 5). Overall, our results support the hypothesis that the structural spinal organization of MN pools embeds the musculoskeletal organization. We demonstrated that the MN pools innervating synergistic muscles were closely located in the spinal cord, and they were adjacent to the MN pools innervating antagonistic muscles (Fig. 6). Such spatial organization of the MN pools can support the co-activation of synergistic muscles they innervate and the reciprocal activation of antagonistic muscles they innervate. The significant reflex contribution to the co-activation of extensor muscles to support weight bearing during locomotion[29–31] is likely facilitated by the close proximity of MN pools innervating these synergistic muscles. This may be an important mechanism for supporting the arm against gravity. Moreover, the reciprocal activation of muscles during locomotion[18,30,32,33] is likely facilitated by the proximity of the MN pools innervating antagonistic muscle groups that have opposing actions in stance and swing phases of gait. In contrast, the observed spatial relationship between MN pools may be less favorable for creating co-contraction of antagonistic muscles, supporting the idea that higher-level neural signals are in control of this action[34,35].

There are important differences between the human and macaque musculoskeletal anatomy that may contribute to the noise in our data. For example, macaques have a limited range of motion at their shoulder due to a different scapular orientation, and are digitigrade quadrupedal animals, which requires hyper-extension of the metacarpophalangeal joints[36]. The added function of using their forelimbs for locomotion also requires that

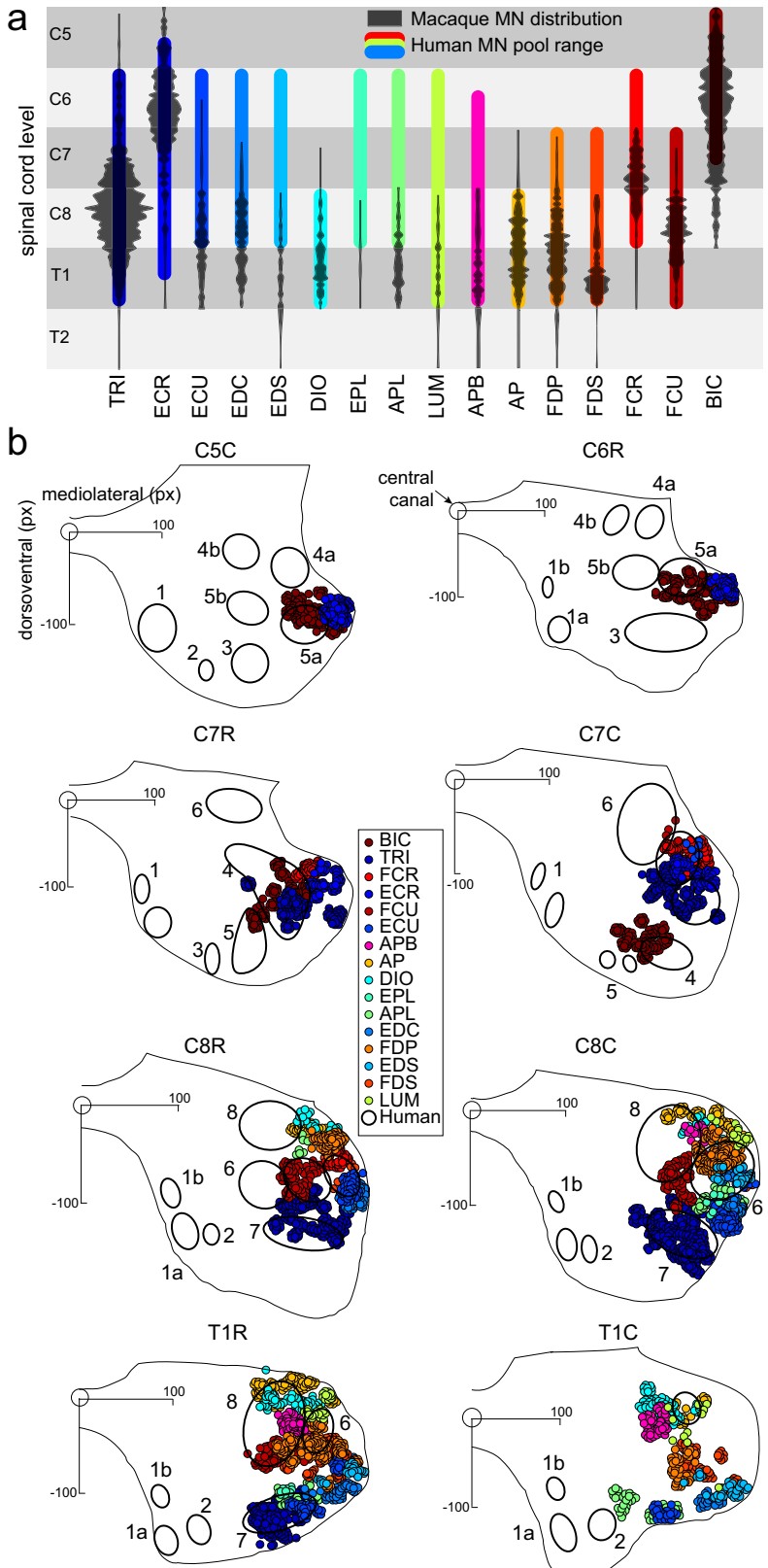

**Fig. 3 Comparison of MN locations between the macaque and human spinal cords. a** Gray violin plots show the distribution of MNs from Fig. 1 along the rostrocaudal direction across spinal segments. Colored bars show the rostrocaudal ranges of corresponding MN pools from a human anatomy textbook[23]. **b** Circles (size no to scale) show MN locations from Fig. 1 in the transverse plane per spinal segment from the macaque model. Gray matter outlines aligned on the central canal at the origin of the axes are from the macaque model. Black open ovals show the distributions of unidentified MN pools adapted from human staining studies summarized and numbered as in ref. [20].

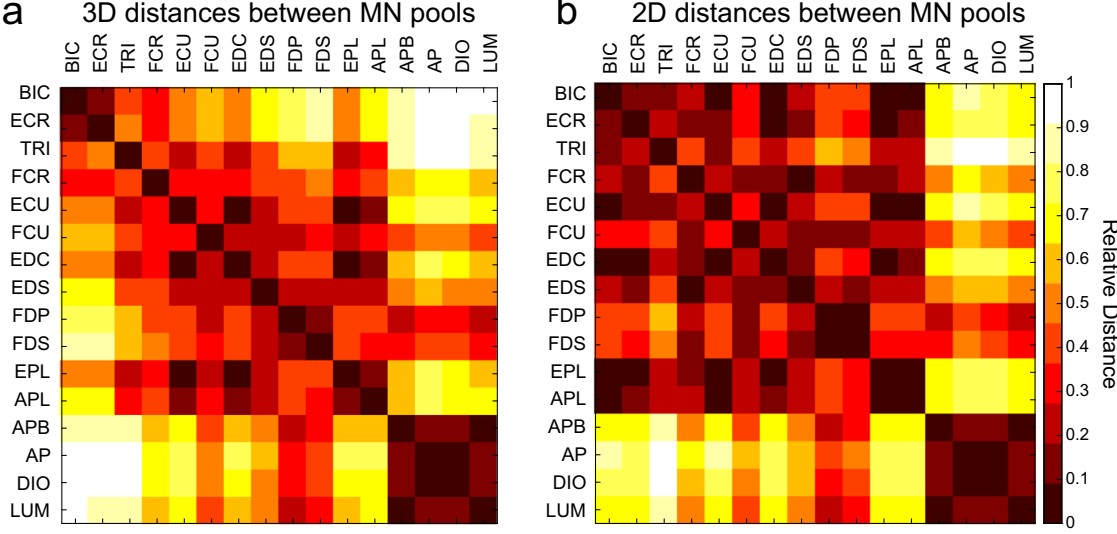

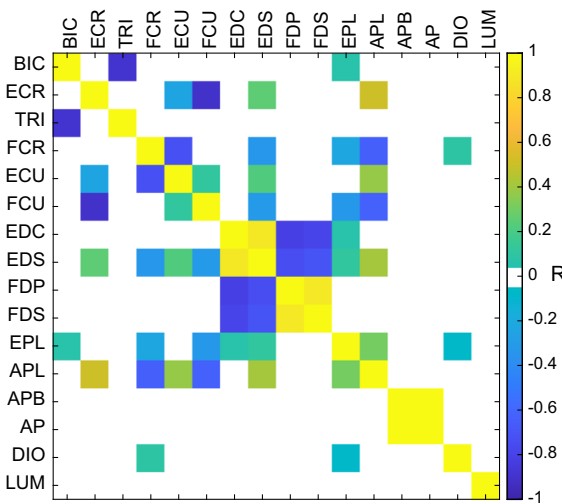

**Fig. 4 Anatomical distances between MN pools and the length correlations between muscles they innervate.** Correlation matrix of the relative distances between the centers of MN pools in all dimensions (**a**) and only in the transverse plane with rostrocaudal distribution excluded (**b**) calculated using the model shown in Fig. 1. Distances were calculated between the centers of MN pools for each pair of MN pools relative to the maximum distance across all pairs. Black indicates short distance, white indicates long distance. **c** Matrix of Pearson correlation coefficients (R) between muscle length changes evaluated across the full range of motion of the macaque forelimb in Fig. 3a only including the muscles whose MN pools were labeled by ref. [24]. Yellow indicates agonistic actions of muscle lengths changing together, i.e., lengthening and shortening together; blue indicates antagonistic actions of reciprocally changing muscle lengths, i.e., lengthening while the other muscle is shortening. Muscles included here are those whose motoneurons are also included in the spinal cord model.

their upper limb musculature can bear weight in a manner similar to their lower limb, unlike humans. Anatomical differences between species (Fig. 2) may result from these functional differences in how the forelimbs are used. For example, homologous trunk and shoulder musculature acting on the front limb of macaques exhibit different origin and insertion points. Additionally, macaques typically have two heads of the coracobrachialis and a single head of the flexor digitorum profundus, while humans have a single coracobrachialis and the flexor digitorum profundus is accompanied by the flexor pollicis longus. Macaques also have the additional dorsoepitrochlearis muscle, which functions similarly to the triceps brachii[37], but has actually found to limit the range of shoulder abduction in humans with this supernumerary muscle present[38]. Despite these anatomical differences, we use macaques for comparison here due to their ability to perform similar movements to humans, for example, the

fractionated finger movements, and the existence of direct corticomotoneuronal projections unlike other species[39–41]. Despite these differences in muscle anatomy, biomechanics, and function, the agonistic/antagonistic relationships between muscles in both species were remarkably similar (Fig. 5). This supports the idea of the conservation of anatomical relationships across species, so that evolutionarily older neural and muscle anatomical structures are adapted to perform new functions.

Our findings further support the idea that the anatomical organization of the spinal cord simplifies the control of movement. Early studies in spinalized animals have shown that microstimulation inside the spinal cord gray matter produces contractions of multiple muscles, which cause converging forces toward a subset of postures[42]. The idea then emerged that muscles can be controlled in groups with fewer control signals of varying spatial and temporal patterns that can create a large

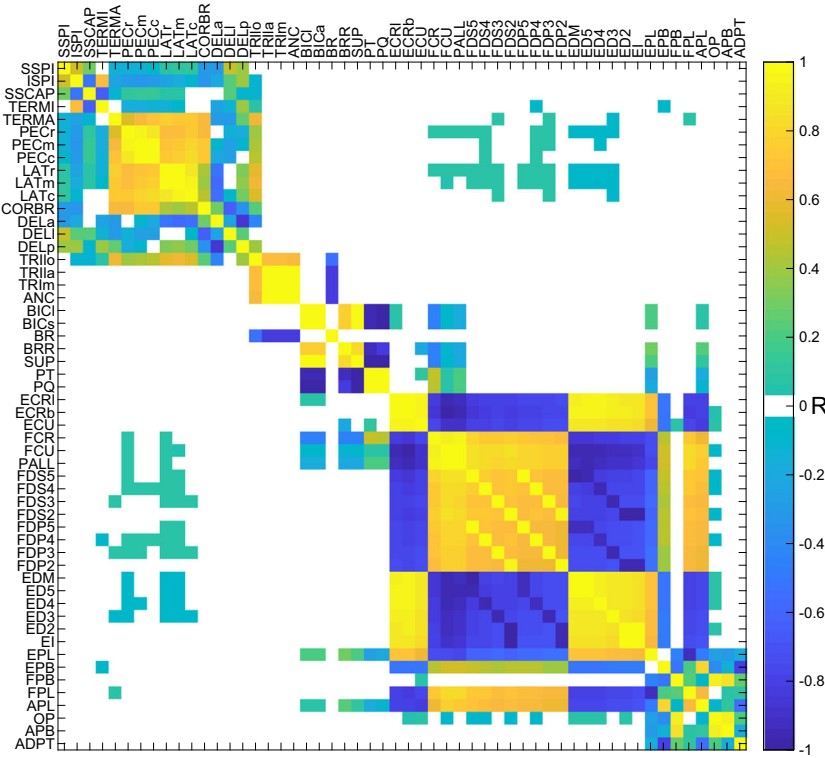

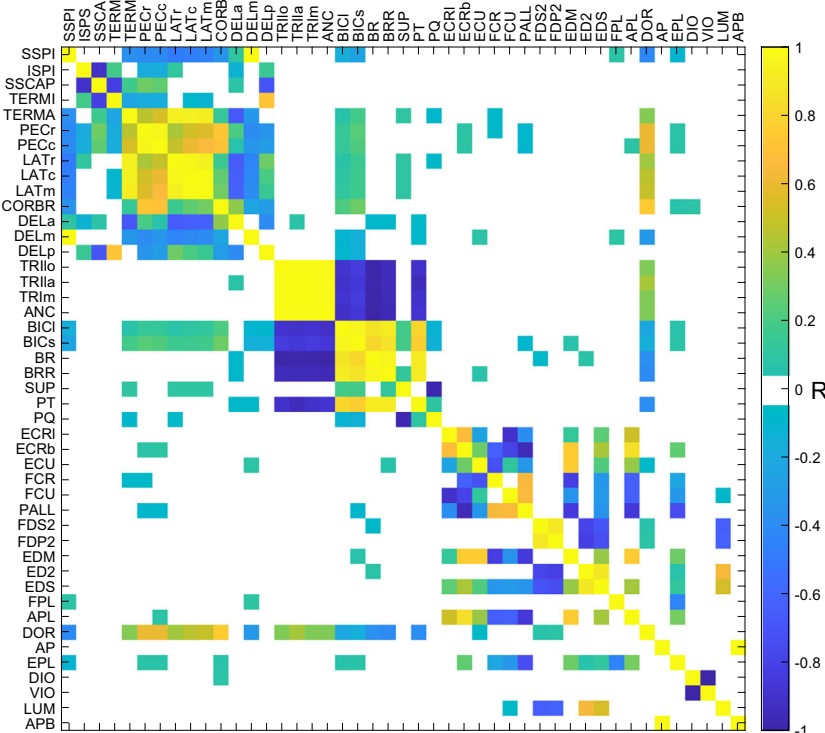

**Fig. 5 The anatomical relationships between upper limb muscles of human and macaque.** Matrices of Pearson correlation coefficients (R) between muscle length changes evaluated across the full range of motion of the OpenSim models of the human arm (**a**) and macaque forelimb (**b**) shown in Fig. 2. Yellow indicates agonistic actions; blue indicates antagonistic actions as in Fig. 4c. Muscle lengths are used here to compare the functional relationships between the upper extremity muscles of human and macaque musculoskeletal models.

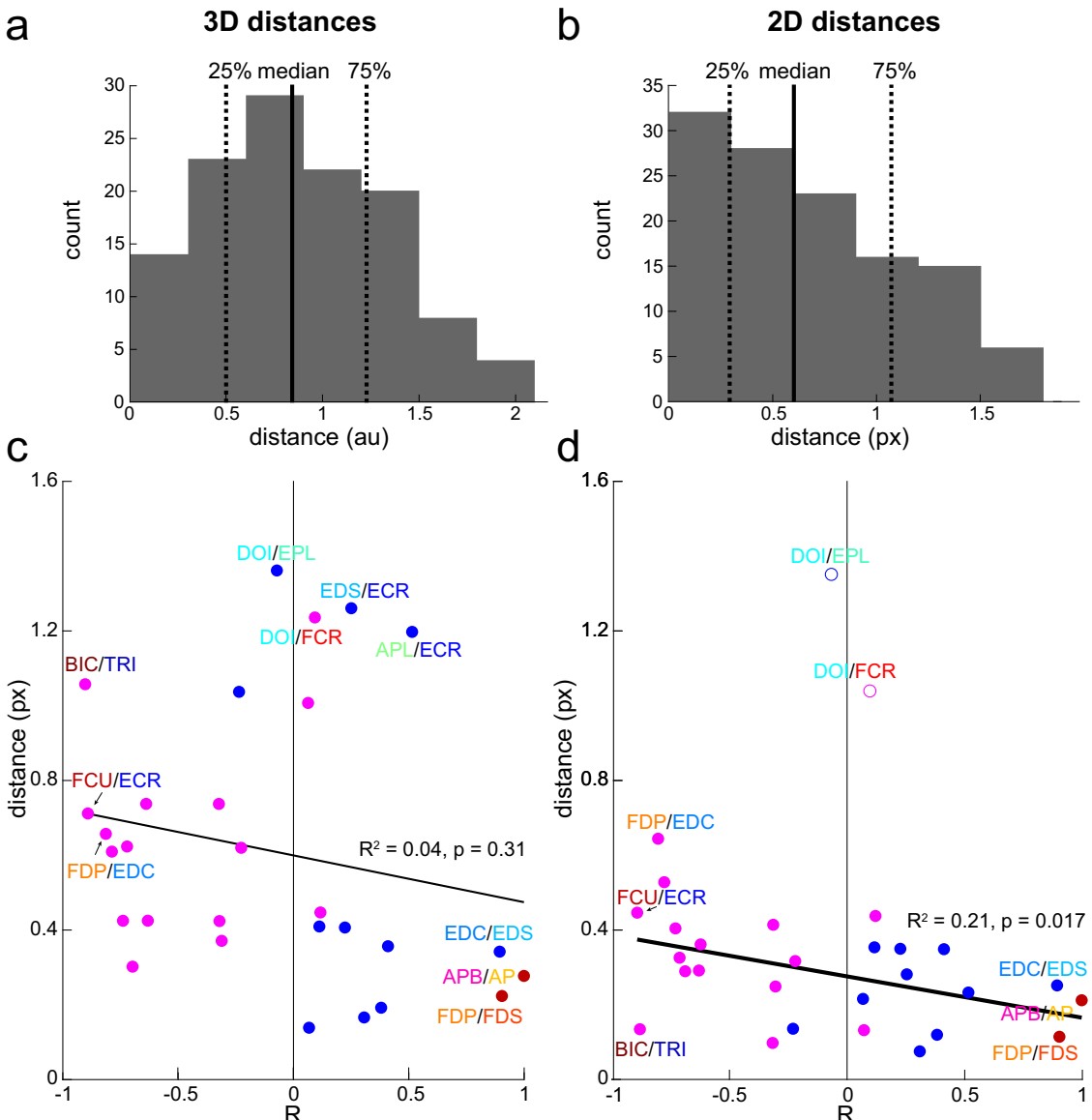

**Fig. 6 Comparative analysis of the spatial arrangement of MN pools and the anatomical relationships between muscles they innervate. a** Histogram of 3D distances between MN pools. Vertical solid and dashed lines show the median and interquartile ranges, respectively. **b** Histogram of 2D distances in the transverse plane between MN pools. Vertical solid and dashed lines show the median and interquartile ranges, respectively. **c** 3D distances between MN pools vs. Pearson correlation coefficients (R) between lengths of muscles they innervate. Blue and red dots show distanced between MN pools that innervate muscles with positively correlated lengths illustrating extensor and flexor synergies, respectively. Magenta dots show distances between MN pools that innervate muscles with negatively correlated lengths illustrating antagonists. **d** The same as in **c**, but for 2D distances. Outliers are shown as open circles.

repertoire of movements[43]. There is also a parallel line of investigation of the spinal control of locomotion by the central pattern generator (CPG)[5]. Evidence supports the importance of CPG in human sensorimotor control[44–46]. There is evidence that the different spatiotemporal dynamics of the CPG and its interaction with the body through afferent feedback can generate different behaviors. This can be seen in neuromechanical studies with lampreys where varying the spatiotemporal patterns of tonic excitation to CPG and its propagation along the spinal segments can produce swimming in either forward or backward direction[47] and change the speed, direction and type of gait[48,49]. Some of the spatiotemporal dynamics of the CPG during gait has been shown to be reflected in the anatomical distribution of the MN pools innervating hindlimb muscles of the cat[18], supporting the idea of the somatotopic "map" of a hindlimb[50]. Our results show that the

MN pools innervating muscles that perform synergistic and reciprocal action are co-localized in the spinal cord. Nearby MN pools are also likely to share the proprioceptive feedback from the muscles they innervate as has been shown for cat knee extensor muscles and their afferents[29]. Similarly, nearby MN pools are likely to receive common descending signals from propriospinal interneurons[51] that can explain observations of the convergent force fields that may serve as building blocks for more complex motor control[42,52]. Furthermore, nearby MN pools are likely to receive common descending signals from the somatotopically organized motor cortex that can explain observations of the complex movements in different areas of workspace produced as a result of the stimulation of the primary and premotor cortex[53]. Our results suggest that the somatotopic organization of MN pools may play an important role in simplifying the sensorimotor

control of the body through the embedding of the musculoskeletal anatomy.

Recent work has shown that it is possible to restore trunk and leg motor functions using biomimetic epidural electrical stimulation of the dorsal roots below the lesion caused by a spinal cord injury[54]. Our results suggest that a similar approach may work for restoring complex movements of the arm after spinal cord injury. A proof of concept study in macaques have shown that it is possible to evoke selective muscle responses by stimulating cervical dorsal roots[55]. Notably, more rostral stimulation in this study evoked responses in BIC and ECR, while more caudal stimulation evoked responses in FDS, EDC, and APB. The rostrocaudal distribution of MN pools innervating these muscles is similar to that shown by our model (Fig. 3a). These observations further support the validity of our model and indicate potential future uses of it for designing next generation neuroprosthetics.

## Methods

**Model of the spatial organization of motoneuron pools.** Locations of motoneuronal cell bodies innervating macaque forelimb muscles were acquired from a study utilizing retrograde transport of horseradish peroxidase in 10 *macaque mulatta* and 4 *macaque fascicularis* monkeys[24]. Images of spinal cord segments from the rostral and caudal parts of the 5th cervical segment (C5) through the 1st thoracic segment (T1) were scanned, digitized using Adobe Illustrator (Adobe Inc) and saved in the vector graphic SVG format. All data processing and analyses were done in MATLAB (MathWorks, Natick, MA). Metadata from the SVG files representing cell body locations in the two orthogonal planes (mediolateral and dorsoventral) and the segmental outlines of white and gray matter were used to generate a three-dimensional (3D) model, similar to our previous work[18]. The metadata contained planar coordinates in pixel units, points on a Bezier curve for gray matter and central canal outlines in pixel units, and colors representing different MN pools and outlines. Each spinal cord section was imported from a separate file. The cell body locations and gray matter outlines defined coordinates in the mediolateral and dorsoventral directions along cartesian axes X and Y, respectively, relative to the central canal. Coordinates of the central canal center were subtracted from all coordinates. To combine data for all MN pools into one model, one representative set of transverse gray matter outlines (*macaque fascicularis* subject 79-1 from ref. [24]) was selected as the standard in our model. It was used for the spatial normalization of both outlines and MN coordinates across subjects along the dorsoventral and mediolateral axes. The X coordinate was inverted for some MN pools to project the data onto the same side of the spinal cord within the corresponding section. The rostral section of the 5th cervical segment (C5R), the rostral and caudal sections of the 2nd thoracic segment (T2C and T2R respectively) were not included in the manuscript figures[24]. Therefore, we utilized gray matter and central canal outlines from adjacent sections.

To define the rostrocaudal coordinates of MNs, gray matter outlines, and central canal (along the Z axis) a width of a representative gray matter outline from an anatomical atlas was used[56]. The distance from the central canal to the furthest point on the ventral horn of the C8 gray matter outline was measured in mm. The average length of each spinal segment was approximated from the rostrocaudal distribution histogram for C8 from experimental measurements[24], in which each bar was 250 mm. There were 20 bars in the C8 segment representing length of about 5 mm. The ratio between this segment length and the gray matter distance was then used to calculate the segment lengths in pixels and set the Z coordinates of MNs, the gray matter outlines, and the central canal outlines in the model.

The digitization of MNs from published bitmap images of spinal cord segment was challenged by image resolution, color representation, and the superposition of cells in the plane of each section. More digitized MNs could happen due to the mislabeling of irregular shapes in pixelated images and fewer digitized MNs could happen due to the omission of some images of the transverse sections in the manuscript figures. To mitigate this, additional data from the published histograms of the quantity of MNs in each segment was utilized[24]. The total number of MNs was counted for each half of the segment (rostral and caudal) from the histograms and compared to the number of MNs obtained from images of the corresponding transverse sections. If the number of MNs in the model was higher than the number of MNs reported in the histogram for the corresponding pool and segment half, then MN indices for that MN pool and segment were randomly selected for removal. This was compensated by adding missing MNs in accordance with the spatial distribution of the given MN pool and segment. The mean position and standard deviation within transverse sections and the range across spinal segments were held constant for each MN pool. The same process was used for inferring MN locations in the missing C5R, T2R, and T2C sections (Fig. 1b).

To analyze the relative anatomical organization of MN pools, first their centers were calculated by averaging all cell coordinates in each pool. The resampling of MN locations described in the previous paragraph ensured that the mean position and standard deviations matched the published data within the transverse plane of each section. The rostrocaudal dimension was converted into the same pixel units based on the published section thicknesses as described above. Therefore, the centers of the MN pools captured the central tendency of the distribution of MN cell bodies within the gray matter of the spinal cord. Euclidean distances were calculated between each MN pool center defined as the mean along each coordinate. Unfortunately, the rostrocaudal distribution in our sample of MN pools was limited, with most centers of MN pools located in adjacent C8 and T1 segments (Fig. 1c). This underrepresented the overall rostrocaudal distribution of MN pools innervating muscles of the upper limb and limited the range of Euclidian distances between MN pools calculated in 3D. Therefore, the Euclidian distances were also calculated separately in the transverse plane, with the rostrocaudal distribution excluded to control for the potential noise it introduced in our analysis. These distances between MN pool centers are referred to in Results as 3D and 2D distances, respectively.

**Musculoskeletal models of upper extremity.** We used open-source simulation software, OpenSim (version 4.1, Stanford University, Stanford, CA, USA), to model the musculoskeletal anatomy of both the macaque and human right upper limb. All muscles acting at the shoulder, elbow, and wrist joints were analyzed in both species, in addition to the extrinsic thumb muscles (Fig. 2). The macaque musculoskeletal model developed by Chan and Moran[57] was modified to add forearm muscles innervated by the MN pools included in the model described above. The model originally contained the following muscles: anconeus (ANC), adductor pollicis (AP), abductor pollicis brevis (APB) and longus (APL), biceps (BICl and BICs for long and short heads respectively), brachialis (BR), brachioradialis (BRR), coracobrachialis (CORBR), deltoid (DELa, DELm, and DELp for anterior, medial, and posterior heads respectively), dorsal interosseous (DIO), dorsoepitrochlearis (DOR, also known as latissimocondyloideus), extensor carpi radialis (ECRl and ECRb for longus and brevis respectively), extensor carpi ulnaris (ECU), extensor digitorum (ED, 2–5 digits), extensor digiti minimi (EDM), extensor digiti secundi proprius (EDS), extensor indicis (EI), extensor pollicis longus (EPL) and flexor carpi radialis (FCR). The following eight forelimb muscles have been added or modified based on anatomical data[58]: (*adductor pollicis (AP*),

dorsal interossei (DIO), extensor digitorum (EDS for superficialis & EDC for communis), extensor pollicis longus (EPL), flexor digitorum profundus (FDP), flexor digitorum superficialis (FDS), lumbricals (LUM), and ventral interossei (VIO) (Fig. 2a). Muscles with multiple heads and/or points of insertion were modeled as separate musculotendon actuators. For example, biceps (BIC) was modeled as two muscles, BIC long and BIC short. Similarly, the extensor digitorum was modeled as 4 muscles, sharing a similar origin, but inserting on each corresponding distal phalange of digits 25 (index through pinkie). To simulate more accurately the length changes of the additional muscles in different postures and during movement, we extended the model to include the carpometacarpal (CMC), metacarpophalangeal (MCP), and interphalangeal (IP) joints of the digits, increasing the total number of degrees of freedom (DOFs) in the macaque model to 27. The carpometacarpal joint of the thumb and the metacarpophalangeal joints of digits 2–4 were modeled with 2 DOFs, representing flexion/extension and abduction/adduction around the x- and z-axes, respectively. Interphalangeal joints were modeled as a single DOF representing flexion/extension about the axes.

A detailed model of the human arm was used to assess musculoskeletal differences between macaque and human musculoskeletal anatomy. The original model developed by Saul et al.[27] was previously expanded to include 23 DOFs and 52 musculotendinous actuators representing all major joints and 33 major muscles of the human arm[25] (Fig. 2b). The moment arms and maximal muscle forces were validated against published measurements[59]. Only the DOFs spanned by muscles innervated by the MN pools modeled above (Fig. 1) were included in the comparative analysis of muscle lengths. These DOFs were elbow flexion/extension, hand pronation/supination, wrist joint flexion/extension and abduction/adduction, CMC joint flexion/extension and abduction/adduction, MCP joints 1-5 flexion/extension, and all IP joints 1-5 flexion/extension.

To analyze the anatomical organization of the musculoskeletal systems, musculotendon lengths (muscle lengths) were calculated for each forelimb posture across the physiological range of motion within the model using MATLAB pipeline tools for OpenSim. The minimum and maximum joint angles were predefined[60] and postures for muscle lengths calculations were selected by increasing each joint angle in 20% increments. Then, a correlation matrix between lengths of muscle pairs across all postures was then computed as in Gritsenko et al.[25]. Positive correlations indicate a synergistic action of muscles, while negative correlations indicate antagonistic action. The correlations between muscle pairs whose MN pools were located closer vs. further apart were compared as described in the Statistics and reproducibility section.

**Statistics and reproducibility**. The 3D and 2D distances between MN pool centers that innervated muscles with significant relationships between their length were selected for statistical analysis in MATLAB. Outliers were removed using rmoutliers function with the default median method, which defined the outliers as elements more than three scaled $D$ from the median. $D$ was calculated as follows:

$$D = \frac{median(|d - \bar{d}|)}{\sqrt{2 \cdot erfcinv\left(\frac{3}{2}\right)}},$$

where erfinv is inverse complementary error function, $d$ are distances between MN pools and $\bar{d}$ is the median distance. Regressions were fitted between the 3D and 2D distances and the corresponding significant Pearson correlation coefficients (R) from muscle length analysis described above using the regress function. Additionally, the distances associated with positive and negative R were compared using a one-tailed

Kolmogorov–Smirnov test using the kstest2 function. The null hypothesis was that the distances come from a population with the same distribution, the alternative hypothesis was that the cumulative distribution function of the distances with negative correlations between the lengths of muscles they innervate is larger than those with positive correlations between the lengths of muscles they innervate. Familywise error for conducting two tests was addressed using Bonferroni adjustment[61], alpha was set to 0.025.

**Reporting summary**. Further information on research design is available in the Nature Portfolio Reporting Summary linked to this article.

## Data availability
The motoneuron coordinates and derivative data used for statistical analysis, such as distances between MN pools and the correlation values between the lengths of muscles they innervate, are shared online[62].

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

## Acknowledgements

We would like to thank Ariel Thomas and Brian Tomblin for digitizing data and Mark Kozy for scripting help. V.G. and S.Y. were supported by NIGMS grants P20GM109098 and P30GM103503. R.I.T. was supported by a fellowship from NIGMS T32 AG052375. This work was supported in part by Office of the Assistant Secretary of Defense for Health Affairs through the Restoring Warfighters with Neuromusculoskeletal Injuries Research Program (RESTORE) under Award No. W81XWH-21-1-0138. Opinions, interpretations, conclusions, and recommendations are those of the author and are not necessarily endorsed by the Department of Defense.

## Author contributions

Conceived and designed the study: V.G. and S.Y.; Data digitization: R.I.T; Model development: R.I.T, V.G., and S.Y.; Analyzed and interpreted the data: R.I.T, V.G., and S.Y.; Wrote the manuscript: R.I.T, V.G., and S.Y.; All co-authors have read and edited the manuscript and agree with its content.

## Competing interests

The authors declare no competing interests.
