## [Peer Review File · Communications Biology]

Reviewers' comments:

Reviewer #1 (Remarks to the Author):

Taitano et al describe a largely theoretical study that attempts to link the spinal cord ventral horn distances between motor neuron pools and their functional anatomy in terms of muscle contraction length changes during various locomotor activities. The paper takes previously published positions of motor neuron pools of the macaque and of humans and also makes use of a validated model of the forelimb of primates.

Whilst this study is likely to be of interest to researchers of locomotor behaviour and motor pool physiology, I think that the manuscript needs some additional work to improve clarity and scope of the study.

For example, the length of the results section is really far too short to allow an in-depth presentation of the results. Running at around 1000 words in total, for a manuscript of 6 figures, the results is just too brief to be easily understandable or to give sufficient detail for the reader to assess the manuscript. Figure 2, for example, is only mentioned in the materials and methods section and not all of the muscles studied in the manuscript are annotated. Additionally, there is very little detail on the model used and the variation in limb position that were included in the analysis.

Further, in figure 3, there appear to be only small overlaps between the location of motor pools in the macaque and the human study mentioned; many, if not most of the putative human motor pools do not overlap with the experimental locations of the macaque pools. This appears to be glossed over by the authors.

In figure 4 C, I think it would be more accurate to label this part of the figure as a correlation in muscle length-changes rather than how it is labelled as "mechanical coupling between muscles"

I also think that further discussion on the lack of a correlation between 3D distances between pools and the analysis of correlation of muscle length changes needs to be strengthened. This, I think, is particularly important given that the correlation of 2D distances, whilst statistically significant, is not terribly high at an R squared of 0.29.

There are also some studies by George Romanes that I think are of strong significance to this manuscript that I think should be cited and discussed throughout the manuscript. Many of these citations can be found in a historical perspective on motor pool position that was published by Jessell et al "Motor Neurons and the Sense of Place", *Neuron*, Volume 72, Issue 3, 2011, Pages 419-424, ISSN 0896-6273, <https://doi.org/10.1016/j.neuron.2011.10.021>.

Another useful source might be McHanwell and Biscoe, 1981- cited in the above.

Romanes was the first to suggest a relationship between position of motor neuron pools in the spinal cord and the position of muscles in the limb. This seems to be important to discuss in the current study. There is also a view that central position of motor neuron pools may be more related to the importance of proprioceptive synaptic input to motor pools, rather than a peripheral link. I think this would also be important for the current authors to consider and discuss in this manuscript.

Taken together, whilst I think this a potentially interesting study, I think that a major revision in the text, to substantially increase discussion of the above, would be important for significance and readability.

Reviewer #2 (Remarks to the Author):

The manuscript by Gritsenko and colleagues investigates the relationship between the spatial organization of motor neuron pools and the organization of corresponding muscle groups. The authors utilize pre-existing retrograde tracing data from macaque to create a 3-dimensional model of forearm motor pool organization and relate that to functional musculoskeletal anatomy. While the authors perform a thorough and comprehensive analysis, the significance and novelty of the study, as well as the broad relevance of the underlying proposed principles is somewhat unclear.

1) The authors utilize the macaque musculoskeletal system to infer functional relationships between motor neurons and muscles, as a closely-related model for humans. While comparisons between human and macaque motor pools and muscle anatomies are shown, there is no quantitation of the similarities and differences. The authors state that both rostrocaudal and transverse coordinates overlap for human and macaque motor pools, but there are clear differences in the data shown in fig.3, making it difficult to appreciate whether the findings for macaque (presumably-this is not clarified in the figure legend or results but it appears to be so from the abstract) in figure 6 also hold true for human.

2) The authors briefly mention that motor neuron pool organization in macaque is similar to cat and human. The location of motor pools has been mapped in other species as well, such as mouse and chick, so it would be interesting to see whether the same relationships between motor pool and muscle organization are evolutionary conserved.

3) The final linear regression analysis done for macaque in fig.6 indicates a correlation of the spatial distribution of MN pools in the transverse, but not rostrocaudal, plane with muscle functional relationships, and only for nearby MN pools. The significance of this finding is not clear, as the authors also comment in the discussion that differential rostrocaudal stimulation is sufficient to evoke selective muscle responses.

Minor:

1) Figure 2 is not referred to in the paper.

Reviewer #3 (Remarks to the Author):

This study investigates whether motoneurons with synergies are located nearby each other in the macaque spinal cord. Motoneurons innervating the forelimbs were identified using retrograde labelling and their location in the macaque was compared to the location of unidentified motoneurons in the cervical and upper thoracic segments of the human spinal cord as reported in previous publications. Their main findings suggest that motoneurons innervating the forelimbs of the macaque are organized in such a way to facilitate the activation of synergies. The comparison with human spinal cord suggests that this would also apply to the human spinal cord. The presence of a topographical distribution in the spinal cord to facilitate recruitment of synergies is useful to know to understand the grand scheme of motor control.

The methodology used seems to be done with careful consideration as to how to best compare the macaque data obtained from retrograde labelling with that in the human spinal cord obtained from cresyl violet staining. Adjustments are made for changes in the morphology of the spinal cord in both species.

I only have a few minor suggestions to clarify the methodology:

Line 78: It is not clear how metadata from imaging is converted into exact parameter values in the 3D mode.

Lines 88-91. Please clarify whether you mean section or segment. The published histogram data of the quantity of MNs would likely be based on spinal segment rather than section.

Lines 104-110. A bit confusing here. If there was some resampling done based upon the previous paragraph, how does the resampling influence the averages of coordinates calculated in this paragraph?

Lines 230-233. In this last part of the results, please clarify how you analyzed the regression for the MN pools that were furthest away. Did you perform a separate analysis on MN pools that were distant from each other? Are these the MNs whose distance was greater than the median as the sample whose data is displayed in Figure 6C and D)? If so, I would suggest showing it by adding a Figure 6E and F. Also, would we not expect the reverse relationship in these motor pools that are further away?

Rebuttal letter

Our responses to the editor and reviewers are presented in **bold writing**.

All line numbers cited in our responses refer to the revised manuscript.

Reviewer #1

Taitano et al describe a largely theoretical study that attempts to link the spinal cord ventral horn distances between motor neuron pools and their functional anatomy in terms of muscle contraction length changes during various locomotor activities. The paper takes previously published positions of motor neuron pools of the macaque and of humans and also makes use of a validated model of the forelimb of primates.

Whilst this study is likely to be of interest to researchers of locomotor behaviour and motor pool physiology, I think that the manuscript needs some additional work to improve clarity and scope of the study.

Thank you. We appreciate your helpful feedback and have detailed the changes to our manuscript below in regard to each specific comment.

For example, the length of the results section is really far too short to allow an in-depth presentation of the results. Running at around 1000 words in total, for a manuscript of 6 figures, the results is just too brief to be easily understandable or to give sufficient detail for the reader to assess the manuscript. Figure 2, for example, is only mentioned in the materials and methods section and not all of the muscles studied in the manuscript are annotated. Additionally, there is very little detail on the model used and the variation in limb position that were included in the analysis.

We have added mentions of Figure 2 in Results and updated the figure to include more annotations of muscles that were included in both models and some that were only present in one model. However, the hand muscles overlap a lot, therefore annotating each for them would make the figure difficult to read. More information on the OpenSim model, as well as the range of forelimb postures analyzed, was added to the Materials and Methods section.

We have also expanded and clarified the description of our results and their implications in Results section as indicated by the text in red in redlined manuscript. The statistical analysis has also been expanded, it is now described in the last paragraph of the Results.

Further, in figure 3, there appear to be only small overlaps between the location of motor pools in the macaque and the human study mentioned; many, if not most of the putative human motor pools do not overlap with the experimental locations of the macaque pools. This appears to be glossed over by the authors.

We have expanded our description of these differences in Results now on lines 202-232 and Discussion on lines 334-353. Briefly, we elaborated on species-specific variations that we think may be the cause for the shifts in spinal MN pool distribution in humans versus macaques illustrated in Fig. 3. We have improved the model resolution in rostrocaudal dimension. The rostral section of the 5th cervical segment (C5R), the rostral and caudal sections of the 2nd thoracic segment (T2C and T2R respectively) were not included in the manuscript figures in Jenny and Innukai. Previously, the missing MNs from those sections were inferred using the histograms as described in the Methods. We then included them in the adjacent sections for which the grey matter outlines were included in the publication (C5C and T1C). To ensure that we reproduce more faithfully the MN locations in the rostrocaudal direction, we now assigned separate rostrocaudal coordinates (Z) to place these MNs into the missing segments and used grey matter outlines from the adjacent sections (Fig. 1). We have also improved the visualization of the macaque MN distributions in Fig. 3. It now shows more overlap and how few MNs there are in the non-overlapping segments.

In figure 4 C, I think it would be more accurate to label this part of the figure as a correlation in muscle length-changes rather than how it is labelled as "mechanical coupling between muscles"

Corrected, thank you.

I also think that further discussion on the lack of a correlation between 3D distances between pools and the analysis of correlation of muscle length changes needs to be strengthened. This, I think, is particularly important given that the correlation of 2D distances, whilst statistically significant, is not terribly high at an R squared of 0.29.

In the Results section on lines 256-292, we discuss in more details the results to make a cohesive narrative of how the MN pool locations are related to the anatomical arrangement of the muscles they innervate. Firstly, we point out that there were very few relationships between the lengths of muscles whose MN pools are located far away. Secondly, we point out that there are short distances between MN pools innervating muscles whose muscle lengths are positively correlated. Thirdly, we point out on that there are short distances in the transverse plane between MN pools innervating muscles whose muscle lengths are negatively correlated. Lastly, we describe on lines 294-310 results of two statistical tests of this relationship, 1) regression showing a linear relationship between MN pool distances in the transverse plane and length correlations of muscle they innervate and 2) a Kolmogorov-Smirnov test rejecting the null hypothesis that the MN pool distances that innervate positively and negatively correlated muscles come from the same distribution. Altogether, we conclude that this suggests that the MN pools are positioned to support both the synergistic co-activation and the reciprocal interactions between the muscles they innervate.

There are also some studies by George Romanes that I think are of strong significance to this manuscript that I think should be cited and discussed throughout the manuscript. Many of these citations can be found in a historical perspective on motor pool position that was published by Jessell et al "Motor Neurons and the Sense of Place", Neuron, Volume 72, Issue 3, 2011, Pages

419-424, ISSN 0896-6273, <https://doi.org/10.1016/j.neuron.2011.10.021>. Another useful source might be McHanwell and Biscoe, 1981- cited in the above. Romanes was the first to suggest a relationship between position of motor neuron pools in the spinal cord and the position of muscles in the limb. This seems to be important to discuss in the current study. There is also a view that central position of motor neuron pools may be more related to the importance of proprioceptive synaptic input to motor pools, rather than a peripheral link. I think this would also be important for the current authors to consider and discuss in this manuscript.

Thank you for bringing these papers to our attention. We have added these references in the first paragraph of the Introduction to acknowledge the priority of their idea. We have also pointed out that these ideas have not been tested for the MN pools innervating the upper limb. Furthermore, in Results we now point out that taken together the MN pool distribution supports the idea of embedding the synergistic functions of muscles. However, in the transverse plane the MN pools innervating the antagonistic muscles are also closely located, which is novel. The latter supports the peripheral link idea of afferent-driven reciprocal interactions between muscles. We have expanded on these ideas in the first paragraph of the Discussion.

Reviewer #2

The manuscript by Gritsenko and colleagues investigates the relationship between the spatial organization of motor neuron pools and the organization of corresponding muscle groups. The authors utilize pre-existing retrograde tracing data from macaque to create a 3-dimensional model of forearm motor pool organization and relate that to functional musculoskeletal anatomy. While the authors perform a thorough and comprehensive analysis, the significance and novelty of the study, as well as the broad relevance of the underlying proposed principles is somewhat unclear.

Thank you for your helpful feedback. We have detailed the changes to our manuscript below in regard to each of your specific comments.

1) The authors utilize the macaque musculoskeletal system to infer functional relationships between motor neurons and muscles, as a closely-related model for humans. While comparisons between human and macaque motor pools and muscle anatomies are shown, there is no quantitation of the similarities and differences. The authors state that both rostrocaudal and transverse coordinates overlap for human and macaque motor pools, but there are clear differences in the data shown in fig.3, making it difficult to appreciate whether the findings for macaque (presumably-this is not clarified in the figure legend or results but it appears to be so from the abstract) in figure 6 also hold true for human.

We have expanded our description of these differences in Results now on lines 202-232 and Discussion on lines 334-353. Briefly, we elaborated on species-specific variations that we think may be the cause for the shifts in spinal MN pool distribution in humans versus macaques illustrated in Fig. 3. We have improved the model resolution in rostrocaudal dimension. The rostral section of the 5th cervical segment (C5R), the rostral and caudal sections of the 2nd thoracic segment (T2C and T2R respectively) were not included in the manuscript figures in Jenny and Innukai. Previously, the missing MNs from those sections were inferred using the histograms as described in the Methods. We then included them

in the adjacent sections for which the grey matter outlines were included in the publication (C5C and T1C). To ensure that we reproduce more faithfully the MN locations in the rostrocaudal direction, we now assigned separate rostrocaudal coordinates (Z) to place these MNs into the missing segments and used grey matter outlines from the adjacent sections (Fig. 1). We have also improved the visualization of the macaque MN distributions in Fig. 3. It now shows more overlap and how few MNs there are in the non-overlapping segments.

We have added a section to Discussion on lines 334-353 outlining the species-specific anatomical and functional differences which may cause the differences in spinal MN pool organization between species. However, we expect that results presented in Figure 6 would be similar in humans due to similar relationships in muscle anatomy between humans and macaques (Fig. 5), but additional studies are needed to validate these results once the detailed human MN distribution in the transverse plane becomes available.

2) The authors briefly mention that motor neuron pool organization in macaque is similar to cat and human. The location of motor pools has been mapped in other species as well, such as mouse and chick, so it would be interesting to see whether the same relationships between motor pool and muscle organization are evolutionary conserved.

Thank you for your suggestion. In first paragraph of the Introduction we have added references to neurodevelopmental studies supporting the co-localization of MNs innervating synergistic dorsal and ventral musculature and expanded our rationale. We have added additional references showing that the organization of MN pools innervating the hindlimb/lower limb is generally preserved across vertebrates (Mott and Sherrington, 1895; Romanes, 1951; Toossi et al., 2021; Yakovenko et al., 2002), including humans (Ivanenko et al., 2006; Routal and Pal, 1999; Sharrard, 1964, 1955). We stated that it is not entirely unexpected as the MN locations innervating the lower limb are preserved across species, because the function of the hindlimb/lower limb is largely conserved to perform stereotypical movements such as locomotion, scratching, or swimming. This is not the case for the upper limb. Macaque and human upper limbs can perform similar functions. Therefore, comparing the locations of macaque MNs that innervate upper limb muscles with the anatomical relationships between these muscles in macaques and humans can help us understand the meaning of the spatial organization of the MN pools in the spinal cord.

3) The final linear regression analysis done for macaque in fig.6 indicates a correlation of the spatial distribution of MN pools in the transverse, but not rostrocaudal, plane with muscle functional relationships, and only for nearby MN pools. The significance of this finding is not clear, as the authors also comment in the discussion that differential rostrocaudal stimulation is sufficient to evoke selective muscle responses.

In the Results section on lines 256-292, we discuss in more details the results to make a cohesive narrative of how the MN pool locations are related to the anatomical arrangement of the muscles they innervate. Firstly, we point out that there were very few relationships between the lengths of muscles whose MN pools are located far away. Secondly, we point out that there are short distances between MN pools innervating muscles whose muscle

lengths are positively correlated. Thirdly, we point out on that there are short distances in the transverse plane between MN pools innervating muscles whose muscle lengths are negatively correlated. Lastly, we describe on lines 294-310 results of two statistical tests of this relationship, 1) regression showing a linear relationship between MN pool distances in the transverse plane and length correlations of muscle they innervate and 2) a Kolmogorov-Smirnov test rejecting the null hypothesis that the MN pool distances that innervate positively and negatively correlated muscles come from the same distribution. Altogether, we conclude that this suggests that the MN pools are positioned to support both the synergistic co-activation and the reciprocal interactions between the muscles they innervate.

We have also expanded the description of the significance of our findings in the first paragraph of the Discussion. Briefly, we state that taken together the MN pool distribution supports the co-activation of synergistic muscles they innervate and the reciprocal activation of antagonistic muscles they innervate. However, the observed spatial relationship between MN pools may be less favorable for creating co-contraction of antagonistic muscles, supporting the idea that higher-level neural signals are in control of this action.

Minor:

1) Figure 2 is not referred to in the paper.

Figure 2 is referenced in the Materials & Methods section in 3 places now. We have expanded our discussion of the musculoskeletal models used in this study in Results and reference Fig. 2 twice and Discussion once.

Reviewer #3

This study investigates whether motoneurons with synergies are located nearby each other in the macaque spinal cord. Motoneurons innervating the forelimbs were identified using retrograde labelling and their location in the macaque was compared to the location of unidentified motoneurons in the cervical and upper thoracic segments of the human spinal cord as reported in previous publications. Their main findings suggest that motoneurons innervating the forelimbs of the macaque are organized in such a way to facilitate the activation of synergies. The comparison with human spinal cord suggests that this would also apply to the human spinal cord. The presence of a topographical distribution in the spinal cord to facilitate recruitment of synergies is useful to know to understand the grand scheme of motor control.

The methodology used seems to be done with careful consideration as to how to best compare the macaque data obtained from retrograde labelling with that in the human spinal cord obtained from cresyl violet staining. Adjustments are made for changes in the morphology of the spinal cord in both species.

Thank you for your feedback. We appreciate your review of our work and have addressed your concerns with the following changes to our manuscript.

I only have a few minor suggestions to clarify the methodology:

Line 78: It is not clear how metadata from imaging is converted into exact parameter values in the 3D mode.

We have expanded our description of the digitization of data and metadata processing on lines 83-106.

Lines 88-91. Please clarify whether you mean section or segment. The published histogram data of the quantity of MNs would likely be based on spinal segment rather than section.

Corrected. We have clarified the terminology in the paragraph now on lines 108-121.

Lines 104-110. A bit confusing here. If there was some resampling done based upon the previous paragraph, how does the resampling influence the averages of coordinates calculated in this paragraph?

The resampling was done in such a way as not to change the mean and standard deviations of each MN pool. We have expanded the description of resampling procedures now on lines 108-121.

The rostral section of the 5th cervical segment (C5R), the rostral and caudal sections of the 2nd thoracic segment (T2C and T2R respectively) were not included in the manuscript figures in Jenny and Innukai. Previously, the missing MNs from those sections were inferred using the histograms as described in the Methods. We then included them in the adjacent sections for which the grey matter outlines were included in the publication (C5C and T1C). To ensure that we reproduce more faithfully the MN locations in the rostrocaudal direction, we now assigned separate rostrocaudal coordinates (Z) to place these MNs into the missing segments and used grey matter outlines from the adjacent sections (Fig. 1).

Lines 230-233. In this last part of the results, please clarify how you analyzed the regression for the MN pools that were furthest away. Did you perform a separate analysis on MN pools that were distant from each other? Are these the MNs whose distance was greater than the median as the sample whose data is displayed in Figure 6C and D)? If so, I would suggest showing it by adding a Figure 6E and F. Also, would we not expect the reverse relationship in these motor pools that are further away?

We have now changed the analysis to include all data for the 3D distances shown in Fig. 6A and C. We have applied an outlier removal method as described in Methods on lines 183-199, which removed 2 values from the 2D distances as shown in Fig. 6D.

We have expanded our statistical analyses to include both regressions and Kolmogorov-Smirnov tests. In the Results section on lines 256-292, we discuss in more details the results to make a cohesive narrative of how the MN pool locations are related to the anatomical arrangement of the muscles they innervate. Firstly, we point out that there were very few relationships between the lengths of muscles whose MN pools are located far away. Secondly, we point out that there are short distances between MN pools innervating muscles whose muscle lengths are positively correlated. Thirdly, we point out on that there are short distances in the transverse plane between MN pools innervating muscles whose muscle lengths are negatively correlated. Lastly, we describe on lines 294-310 results of

two statistical tests of this relationship, 1) regression showing a linear relationship between MN pool distances in the transverse plane and length correlations of muscle they innervate and 2) a Kolmogorov-Smirnov test rejecting the null hypothesis that the MN pool distances that innervate positively and negatively correlated muscles come from the same distribution. Altogether, we conclude that this suggests that the MN pools are positioned to support both the synergistic co-activation and the reciprocal interactions between the muscles they innervate.

REVIEWERS' COMMENTS:

Reviewer #2 (Remarks to the Author):

The authors have addressed all my previous concerns.

Reviewer #3 (Remarks to the Author):

The authors have addressed well my comments. There were substantial changes in the text in response to all reviewer comments and I leave it to my fellow reviewers to assess how their comments were addressed.